# Spatially dispersing Yu-Shiba-Rusinov states in the unconventional superconductor FeTe$_{0.55}$Se$_{0.45}$

Damianos Chatzopoulos[1,8], Doohee Cho[1,2,8], Koen M. Bastiaans[1,8], Gorm O. Steffensen[3], Damian Bouwmeester[1,4], Alireza Akbari[5,6], Genda Gu[7], Jens Paaske[3], Brian M. Andersen[3] & Milan P. Allan [1✉]

By using scanning tunneling microscopy (STM) we find and characterize dispersive, energy-symmetric in-gap states in the iron-based superconductor FeTe$_{0.55}$Se$_{0.45}$, a material that exhibits signatures of topological superconductivity, and Majorana bound states at vortex cores or at impurity locations. We use a superconducting STM tip for enhanced energy resolution, which enables us to show that impurity states can be tuned through the Fermi level with varying tip-sample distance. We find that the impurity state is of the Yu-Shiba-Rusinov (YSR) type, and argue that the energy shift is caused by the low superfluid density in FeTe$_{0.55}$Se$_{0.45}$, which allows the electric field of the tip to slightly penetrate the sample. We model the newly introduced tip-gating scenario within the single-impurity Anderson model and find good agreement to the experimental data.

[1] Leiden Institute of Physics, Leiden University, Niels Bohrweg 2, Leiden, CA 2333, The Netherlands. [2] Department of Physics, Yonsei University, Seoul 03722, Republic of Korea. [3] Center for Quantum Devices, Niels Bohr Institute, University of Copenhagen, Universitetsparken 5, Copenhagen Ø 2100, Denmark. [4] Kavli Institute of Nanoscience, Delft University of Technology, Lorentzweg 1, Delft, CJ 2628, Netherlands. [5] Max Planck Institute for the Chemical Physics of Solids, Dresden D-01187, Germany. [6] Max Planck POSTECH Center for Complex Phase Materials, and Department of Physics, POSTECH, Pohang, Gyeongbuk 790-784, Korea. [7] Condensed Matter Physics and Materials Science Department, Brookhaven National Laboratory, Upton, NY 11973, USA. [8] These authors contributed equally: Damianos Chatzopoulos, Doohee Cho, Koen M. Bastiaans. ✉email: allan@physics.leidenuniv.nl

The putative $s_{\pm}$ superconductor $FeTe_{0.55}Se_{0.45}$ is peculiar because it has a low Fermi energy and an unusually low and inhomogeneous superfluid density[1–7]. It has been predicted to host a topological superfluid and Majorana zero-mode states[8–10]. These predictions have been supported by recent experiments: photoemission has discovered Dirac-like dispersion of a surface state[11] while tunneling experiments have concentrated on in-gap states in vortex cores, which have been interpreted as Majorana bound states[12,13] since the low Fermi energy allows to distinguish them from conventional low-energy Caroli-Matricon-de Gennes states[14].

In-gap states have a long history of shining light into the properties of different host materials, and have allowed to bring insight into gap symmetry and structure, symmetry breaking, or the absence of scattering in topological defects, to name a few[15–24]. Impurity bound states have also been investigated in chains or arrays of magnetic impurities on superconducting surfaces where they can lead to Majorana edge-states[25–28]. In the case of $FeTe_{0.55}Se_{0.45}$, zero-bias in-gap resonances have become a primary way to identify Majorana bound states at magnetic impurity sites or in vortex cores. At impurity sites, robust zero-bias peaks have been reported at interstitial iron locations which suggest the presence of Majorana physics[29]. In addition, very recently STM experiments reported signatures of reversibility between magnetic impurity bound states and Majorana zero modes by varying the tip-sample distance on magnetic adatoms[30]. Interestingly, there have also been signatures of spatially varying in-gap impurity states[31,32] which are not yet understood.

Here we report the detection of in-gap states at sub-surface impurities, which are spatially dispersing, i.e., they change energy when moving away from the impurity site by a distance of $\Delta y$. The energy can also be tuned by changing the tip-sample distance ($\Delta d$). We argue that the most likely explanation of our observations involves a magnetic impurity state of the YSR type affected by the electric field of the tip. We show good agreement between our experimental findings and the single impurity Anderson model.

## Results

**Detection of a particle-hole symmetric in-gap state in $FeTe_{0.55}Se_{0.45}$.** We use $FeTe_{0.55}Se_{0.45}$ samples with a critical temperature of $T_C = 14.5$ K. They are cleaved at ~30 K in ultra-high vacuum, and immediately inserted into a modified Unisoku STM at a base temperature of 2.2 K, for preventing surface reconstruction and contamination. To increase the energy resolution, we perform all tunneling experiments using a superconducting tip, made by indenting mechanically ground Pt-Ir tips into a clean Pb(111) surface. With the superconducting tip and to leading order in the tunnel coupling, the current-voltage ($I - V$) characteristic curves are proportional to the convolution of the density of states of Bogoliubov quasiparticles in the tip and the sample

$$I(\mathbf{r}, V) \sim \int D_t(\omega + eV) D_s(\mathbf{r}, \omega)[f(\omega, T) - f(\omega + eV, T)]d\omega, \tag{1}$$

where $D_{s(t)}$ is the density of states of the quasiparticles in the sample (tip), $f(\omega, T)$ is the Fermi-Dirac distribution at temperature $T$ and $e$ is the electron charge. In such a superconducting tunnel junction the coherence peaks in the conductance spectra, $dI/dV(\mathbf{r}, V)$, appear at energies: $\pm (\Delta_t + \Delta_s)$, where $\Delta_{s(t)}$ is the quasiparticle excitation gap of the sample (tip). In addition, the energy resolution is far better than the conventional thermal broadening of ~$3.5 k_B T$ ($k_B$ is the Boltzmann constant) since it is enhanced by the sharpness of the coherence peaks of $D_t$[33–35]. To obtain the intrinsic local density of states (LDOS) of the sample, $D_s(\mathbf{r}, \omega)$, we numerically deconvolute our measured $dI/dV(\mathbf{r}, V)$ spectra while retaining the enhanced energy resolution (for more details see Supplementary Note 1). For this, we use our knowledge of the density of states of the tip with a gap of $\Delta_t = 1.3$ meV from test experiments on the Pb(111) surface using the same tip.

Figure 1a shows a topography of the cleaved surface of $FeTe_{0.55}Se_{0.45}$ obtained with a Pb coated tip (see inset). Brighter (darker) regions correspond to Te (Se) terminated areas of the

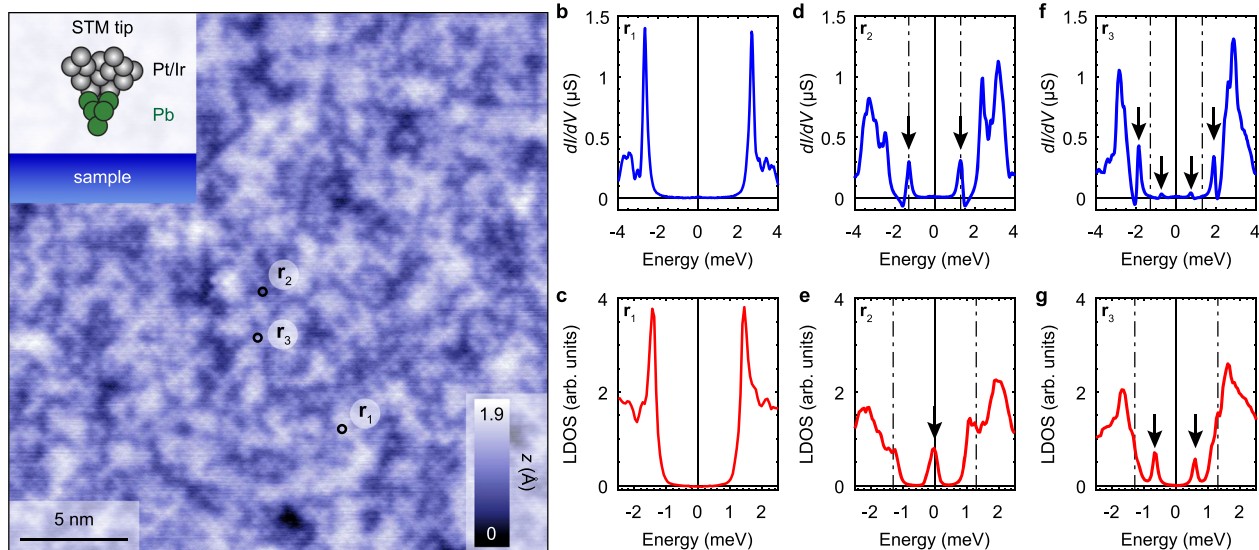

**Fig. 1 Scanning Tunneling Microscopy on FeTe$_{0.55}$Se$_{0.45}$ with a superconducting tip. a** Atomically resolved topographic image ($z$-height map, 25 × 25 nm$^2$) of FeTe$_{0.55}$Se$_{0.45}$ cleaved surface acquired with a Pb coated Pt/Ir tip (see inset) at 2.2 K in ultra-high vacuum. Setup condition: $V_{set} = -8$ mV, $I_{set} = -100$ pA. **b, d, f** Spatially averaged differential conductance spectra in the areas ($r_1$, $r_2$, $r_3$) marked by the black circles in **a**. $r_1$: no in-gap states. $r_2$: two in-gap resonances at ± 1.3 meV. $r_3$: two sets of peaks symmetric in energy around the Fermi level. **c, e, g** Deconvolution of the spectra shown in **b**, **d**, **f**, respectively, provide information about the intrinsic LDOS of the sample in the indicated areas. In $r_2$ a zero-bias impurity state is recovered and in $r_3$ two in-gap states are observed. Setup conditions: **b** $V_{set} = 6$ mV, $I_{set} = 1.2$ nA, **d, f** $V_{set} = 5$ mV, $I_{set} = 2$ nA. Lock-in modulation is $V_{mod} = 30$ $\mu$V peak-to-peak for all measured spectra.

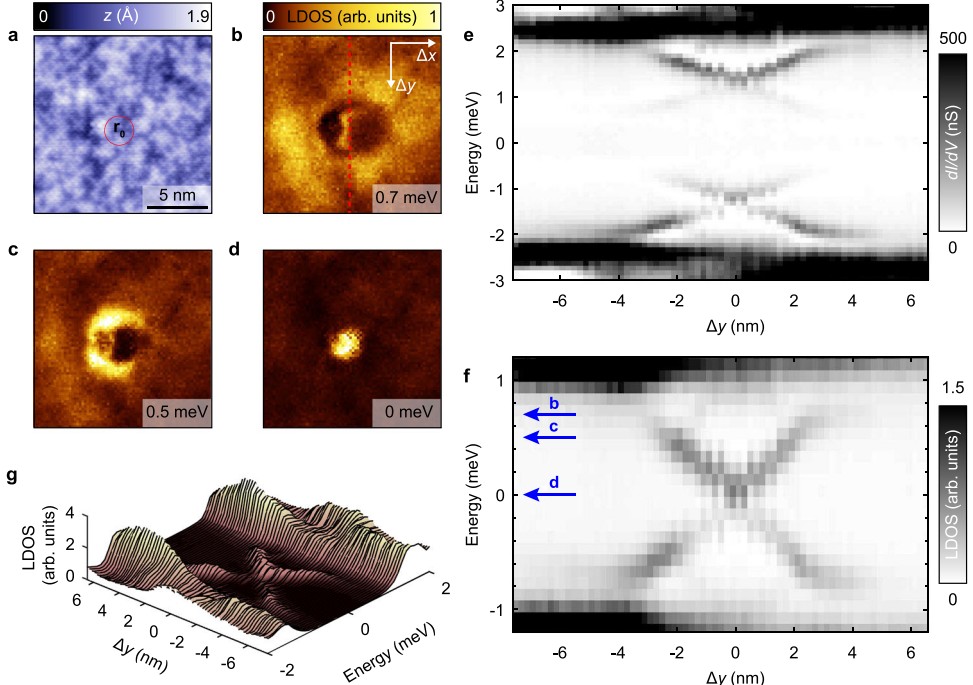

**Fig. 2 X-shaped spatial dispersion of impurity resonances in FeTe$_{0.55}$Se$_{0.45}$. a** Topographic image at the impurity location ($\mathbf{r}_0$ indicates the impurity center). No clear signature of the impurity is observed. Setup conditions: $V_{set} = -8$ mV, $I_{set} = -100$ pA. **b–d** Spatially resolved LDOS maps at different energies obtained by deconvolution of a $dI/dV$ ($\mathbf{r}$, $V$) map in the same field-of-view as in **a**. The energy of each LDOS map is indicated at the bottom right corner. **e** Measured differential conductance intensity plot of a vertical linecut passing through the impurity center $\mathbf{r}_0$ ($\Delta y = 0$ nm). The linecut was taken along the red dashed line in **b**. A crossing of the in-gap resonances at the impurity center is observed. Setup conditions: $V_{set} = -8$ mV, $I_{set} = -1.6$ nA. Lock-in modulation is $V_{mod} = 100$ $\mu$V peak-to-peak, **f** Deconvolution of the measured spectra in **e** shows an X-shaped dispersion of the sub-gap states crossing the Fermi level at the impurity center. The blue arrows indicate the energy of the maps in **b–d**. **g** Series of LDOS spectra depicting the X-shaped spatial dispersion shown in **f**.

cleaved surface which has a tetragonal crystal structure. Our samples exhibit no excess Fe atoms or clusters on the cleaved surface. Spatially resolved scanning tunneling spectroscopy shows that most locations have a flat gap, as shown in Fig. 1b, c. However, when we acquire spectra at specific points indicated by black circles ($\mathbf{r}_2$ and $\mathbf{r}_3$) in Fig. 1a, we find sharp in-gap states. Figure 1d–e and f–g shows such states, both in the raw data as well as in the deconvoluted results. The measured in-gap state is symmetric in energy, i.e., it is visible at $\pm E_{ig}$ (Fig. 1g), or at the Fermi level, $E_{ig} = 0$ (Fig. 1e). In the raw data (before numerical deconvolution) the states are located at energies $\pm(\Delta_t \pm E_{ig})$ (see arrows in Fig. 1d and f) due to the use of the superconducting tip.

**Spatial dispersion of the in-gap state.** In order to characterize the impurity in more detail we acquire a spatially resolved $dI/dV$ ($\mathbf{r}$, $V$) map in the area shown in Fig. 2a. Three energy layers of the deconvoluted map depicting the LDOS variations are shown in Fig. 2b–d. The impurity exhibits a clear ring-shaped feature which eventually becomes a disk with smaller radius at the Fermi level. A spatial line cut profile along the red dashed line shown in Fig. 2b reveals two symmetric resonances around zero energy that extend over ~10 nm in space (Fig. 2e). Importantly, the energies of the in-gap states vary spatially as shown in the spatial cuts (Fig. 2e–g) obtained from the same conductance map. The dispersion of the in-gap states shows an X-shaped profile where the crossing point is indicated with $\mathbf{r}_0$ (Fig. 2a). In more detail, the state is at zero energy at $\mathbf{r}_0$, and then moves away from the Fermi level, before fading out slightly below the gap edge. We will show later that the character of this dispersion is dependent on the tip-sample distance, and that there can also be zero or two crossing points. By inspecting the topography at $\mathbf{r}_0$ we find no signature of

irregularities, which points towards a sub-surface impurity defect as the cause of the observed in-gap peaks in the spectra. We note that these impurities are sparse; we found a total of 5 in a $45 \times 45$ nm$^2$ field-of-view. These all show the same characteristic dispersions, but the X point is estimated at different tip heights. For details, see Supplementary Note 7. Similar observations have been reported previously on FeTe$_{0.55}$Se$_{0.45}$, but without a clear energy cross at the Fermi level[31,32].

**YSR impurity states.** Our observations are reminiscent of YSR states caused by magnetic impurities in conventional superconductors[35–39]. When a single magnetic impurity is coupled to a superconductor with energy gap $\Delta$ via an exchange coupling $J$ then the ground state of the many-body system depends on the interplay between superconductivity and the Kondo effect (described by the Kondo temperature $T_K$). For $\Delta \gtrsim k_B T_K$ the superconducting ground state prevails (unscreened impurity) whereas for $\Delta < k_B T_K$ the Kondo ground state dominates (screened impurity). In each case, quasiparticle excitations above the ground state give rise to resonances symmetrically around the Fermi level inside the superconducting gap. In an STM experiment, this results in peaks in the conductance spectrum at the energy of the two YSR excitations which is determined by the product $v_F J S$, where $S$ is the impurity spin and $v_F$ the normal state density of states in the superconducting host (FeTe$_{0.55}$Se$_{0.45}$ in our case).

It is important to note that the $s_\pm$ symmetry of the order parameter in FeTe$_{0.55}$Se$_{0.45}$ can lead to a very similar phenomenology between magnetic and potential scatterers. While in conventional $s$-wave superconductors, magnetic impurities are required to create in-gap (YSR) states, in $s_\pm$ superconductors, sub-gap resonances can also occur for non-magnetic scattering

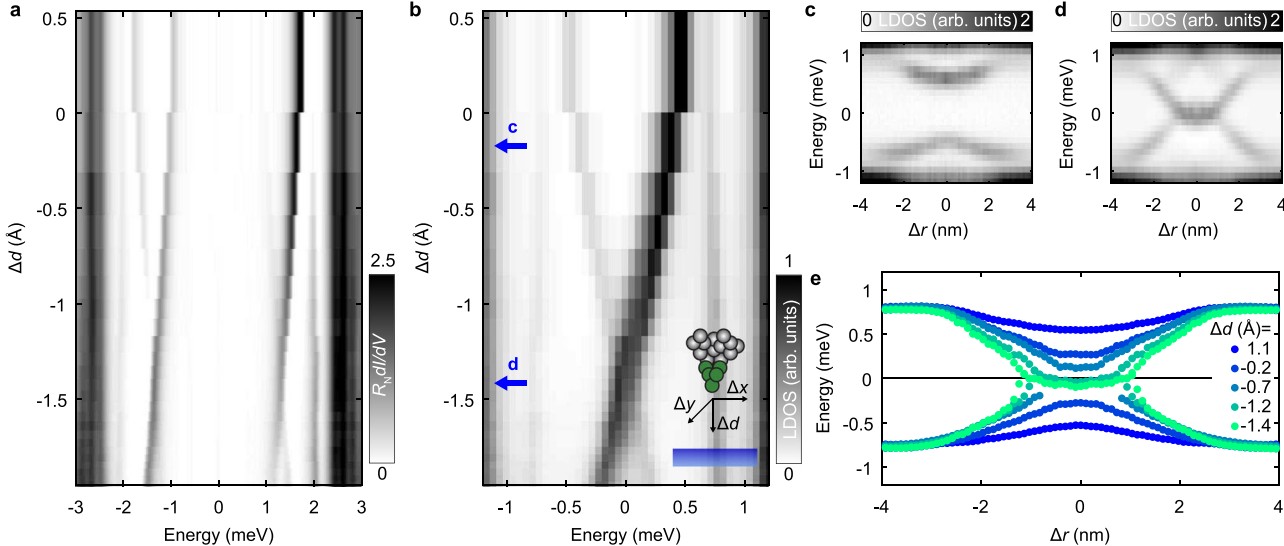

**Fig. 3 Energy dispersion of the sub-gap resonances as a function of the tip-sample distance. a** Conductance intensity plot for varying tip-sample distance ($\Delta d$) normalized by the normal state resistance $R_N$. The in-gap states disperse and cross at the Fermi level. **b** Same as in **a** for the deconvoluted LDOS data. Inset: schematic representation of the tip movement when we vary the tip-sample distance ($\Delta d$) and when the tip scans laterally ($\Delta y$ and $\Delta x$) at a constant height. **c**, **d** Azimuthally-averaged radial profiles at different tip-sample distances indicated by blue arrows in **b**. $\Delta r$, indicates the radial offset $\sqrt{\Delta x^2 + \Delta y^2}$ from the impurity center ($\Delta r = 0$). **e** Energy of the impurity bound state for varying tip-sample distance, extracted by fitting a lorentzian curve in 5 intensity plots (see Supplementary Note 3) similar to **c**, **d**.

centers. This can be shown using different theoretical techniques, including T-matrix method[40,41], Bogoliubov-de Gennes equations[42,43] and Green's functions[44–46] applied to multiband systems with $s_\pm$ symmetry. The similarity of magnetic and potential scatterers makes a distinction between these cases more challenging (but possible, with an external magnetic field[30]). In either case, theory predicts energy-symmetric in-gap states with particle-hole asymmetric intensities.

The X-shaped phenomenology of the in-gap states shown above shares also similarities with bound-states that have been observed in Pb/Co/Si(111) stacks[47], where they have been interpreted as topological[47,48]. However, as we will show here, in our experiments the single point of zero bias is just one particular case of a manifold of dispersions that depend on the tip-sample distance.

**Tuning the energy of the in-gap state with the tip**. Figure 3a shows an intensity plot of a series of spectra above $\mathbf{r}_0$, the location showing the zero-bias impurity state, with changing tip-sample distance (see inset for a schematic). We normalized each spectrum by the normal state resistance $R_N = V_{set}/I_{set}$. In order to reduce the distance, we control the tip in constant current feedback and increase the set-point for the current while keeping the voltage bias constant. In addition, we measure the tip-sample distance relative to the set-point: $V_{set} = 5$ mV, $I_{set} = 0.4$ nA. Strikingly, we observe a shift in the energy of the in-gap state with varying the tip-sample distance $\Delta d$. When the tip is brought closer to the sample surface, the sub-gap resonances shift towards the Fermi energy (Fig. 3b) where they cross and split again. We also point out that there is a strong particle-hole asymmetry in the intensity of the in-gap resonances. It can be clearly seen in Fig. 3b that the relative intensity between the positive (p) and negative (n) resonances ($I_p - I_n$) changes sign after the cross at the Fermi level. Further, we note that the energy shift for varying $\Delta d$ is stronger than the spatial dispersion. This can likely be explained by the tip shape, which can be approximated as a sphere of roughly 20 nm. As the tunneling current falls off exponentially with distance, one always tunnels to the point

closest to the surface. However, the field is algebraic in the distance, and thus a change horizontally has less of an effect than a change vertically, as shown in Supplementary Note 5.

To obtain a more complete picture of the tuning of the in-gap states as we vary $\Delta d$, we measured five $dI/dV(\mathbf{r}, V)$ maps (each at different tip-sample distance) and analyzed azimuthally-averaged radial profiles through the impurity center (Fig. 3c–d shows two of these profiles. See Supplementary Note 3 for the other 3). We extract the energy of the resonances by Lorentzian fits (Fig. 3e), to observe that they cross the Fermi level at the impurity center when being close to the sample. This is the first time that such a crossing has been observed in an unconventional superconductor.

**Microscopic origin**. The important question that arises is: what tunes the impurity resonances that we observe? In previous experiments with magnetic ad-atoms or ad-molecules on conventional superconductors[49,50], it has been shown that the force of the tip changes the coupling between moment and substrate, and that the coupling $J$ and the YSR energy could be tuned in this way. In this case, when the energy crosses the Fermi level at the critical coupling $J_C$, a first-order quantum phase transition between the singlet (screened) and the doublet (unscreened) ground state is expected[49,51]. Very recently, a similar force-based scenario has been reported in different systems involving magnetic ad-atoms on top of superconductors[15], including Fe(Te,Se)[30]. As discussed in Supplementary Note 2, a similar scenario can in principle explain the sub-gap dispersion discovered here. However, as the impurity is not loosely bound on top of the surface in the present case, a movement between a sub-surface impurity and the superconductor due to the tip force as the cause for the tuning, seems unlikely. Therefore, we pursue alternative mechanisms. Motivated by the phenomenology of semiconductors[52] or Mott insulators[53], where the tip can act as a local gate electrode (mediated by the poor screening), we propose a similar gating scenario for YSR states in the present case: the electric field of the tip can tune the energy of the impurity state and thus lead to a dispersing YSR state.

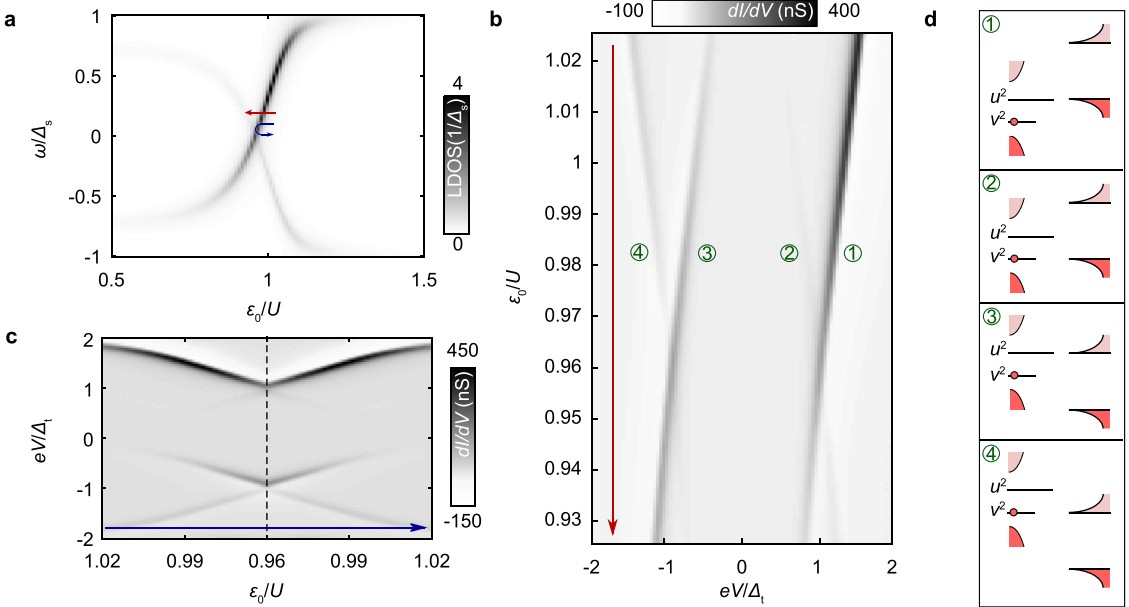

**Fig. 4 Anderson impurity model for Yu-Shiba-Rusinov bound states. a** Local density of states as a function of level energy $\epsilon_0$. The impurity spectral function was calculated within the zero-bandwidth approximation using the Lehmann spectral representation for the retarded Green's function (see Supplementary Note 2). Red and blue lines indicate the two different $\epsilon_0$ sweeps plotted in **b** and **c**, respectively. **b, c** Relaxation dominated tunneling conductance calculated to leading order in the tip-impurity tunneling rate $\Gamma_t$. Labels in **b** refer to processes in **d**. The dashed line in **c** shows the turning point of the blue line in **a**. **d** Guide to the eye for different conductance contributions in **b** and **c**. Processes 2–3 require a finite population of the excited state, in this case supplied by temperature. For all panels we use $U = 20$, $\Gamma_s = 3$, $\Gamma_t = 0.04$, $\gamma = \Gamma_r = 0.035$ and $k_B T = 0.2$, all in units of $\Delta_s$.

First, we note that there can be a significant difference between the work functions of the tip and the sample. Typical work functions are in the range of a few electronvolts, and differences between chemically different materials of the order of an electronvolt are common (see Supplementary Note 5). Hence, it is possible to have a voltage drop between them that is larger than the applied bias. Secondly, the low carrier density in $FeTe_{0.55}Se_{0.45}$ leads to a non-zero screening length giving rise to penetration of the electric field of the tip inside the sample. An estimation of the penetration depth in the sample can be made in the Thomas-Fermi approximation (cf. e.g.[54]). In this framework, the screening length is given by $\lambda_{TF} = (\pi a_0/4k_F)^{1/2}$, where $a_0$ is the Bohr radius and $k_F$ the Fermi wave-vector. Using reported parameters[11,12], this yields $\lambda_{TF} = 0.5$ nm, which is comparable to the inter-layer distance[55]. This implies that in principle, an impurity residing between the topmost layers can be affected by the electric field of the tip. In Supplementary Note 6 we further test this possibility by performing an estimate of the potential shift in the impurity, when the tip-sample distance is varied using a simple model for screening (image charges method) to estimate that the shift is comparable to the charging energy of the impurity. We note that this estimate is approximate, as some key parameters are unknown for $FeTe_{0.55}Se_{0.45}$.

Based on these considerations, we conclude that it is possible that the tip acts as a local gate electrode that influences the energy levels of the impurity, which in turn influences the energy of the in-gap states, as we will demonstrate in the modelling carried out below. By adjusting the tip-sample distance the field penetration is modulated leading to an energy shift of the in-gap resonances. The spatial dependence can be explained similarly: when moving the tip over the impurity location, we change the local electric field, which is at a maximum when the tip is right above the impurity. We emphasize that, similar to experiments on semiconductors and Mott insulators, we expect that the details of the gating process depend on the tip shape Supplementary Note 5.

**Gate-tunable single impurity Anderson model**. We model the sub-gap state as a YSR state arising from the magnetic moment of a sub-surface impurity level, whose energy is effectively gated by the tip-induced electric field. It should be noted that the sub-gap states arising in an $s_\pm$-wave superconductor from a simple non-magnetic impurity can produce a dispersive cross in the in-gap energies as a function of the impurity potential. However, this is only true for a particular range of potentials, and will not generally trace out a single dispersive cross as a function of the impurity strength[42,44]. Therefore, we are led to conclude that the impurity at hand involves a finite magnetic moment. Local impurity-induced magnetic moments may indeed be particularly prominent in correlated systems like FeSe where even non-magnetic disorder, in conjunction with electron interactions, can generate local moments[56]. Because of the magnetic nature of the impurity site, the results of our calculations are qualitatively independent on whether we treat the system as an $s$ or $s_\pm$-wave superconductor. For simplicity, we perform our calculations assuming standard $s$-wave pairing.

The superconducting single impurity Anderson model[57] involves an impurity level $\epsilon_0$ with charging energy $U$ coupled via a tunneling rate $\Gamma_s$ to a superconducting bath with energy gap $\Delta_s$[58–60]. We represent the sample by a simple $s$-wave Bardeen–Cooper–Schrieffer (BCS) superconductor, and use the zero-bandwidth approximation, including only a single spin-degenerate pair of quasiparticles at energy $\Delta_s$[59,61]. We further assume that the gating from the tip changes the impurity level $\epsilon_0$ linearly with distance. We then obtain the YSR states by calculating the local impurity spectral function, $D_I(\omega, \epsilon_0)$, as a function of $\epsilon_0$ (and thus of gating) using the Lehmann representation (see Supplementary Note 2 for details). The result is plotted in Fig. 4a, where the observed crossing of the sub-gap states indicates a change between singlet, and a doublet ground state[62]. From the spectral function we can determine the current using leading-order perturbation theory in the tunnel coupling

connecting the impurity to the tip, $t_t$:

$$I(V) = \frac{e|t_t|^2}{\hbar} \int D_t(\omega + eV, \Delta_t, \gamma_t) D_I(\omega, \epsilon_0)[f(\omega, T) - f(\omega + eV, T)] d\omega,$$

(2)

here $D_t(\omega, \Delta_t, \gamma_t)$ is the spectral function of the superconducting tip with a finite quasiparticle broadening incorporated as a Dynes parameter[63], $\gamma_t$ and $\hbar$ the reduced Planck's constant. A phenomenological relaxation rate, $\Gamma_r$, is incorporated into the Lehmann representation, (see Supplementary Note 2), to construct $D_I(\omega, \epsilon_0)$. This parameter accounts for quasiparticle relaxation of the YSR resonances at $\omega = \pm E_{ig}$. The validity of the expansion in $\Gamma_t = \pi \nu_F |t_t|^2$, which captures only single electron transport and omits Andreev reflections, rests on the assumption that the sub-gap state thermalizes with rate $\Gamma_r$ between each tunneling event. In the opposite limit, $\Gamma_t \gg \Gamma_r$, transport takes place via resonant Andreev reflections, and the sub-gap conductance peaks at $eV = \pm(\Delta_t + E_{ig})$ display a bias asymmetry that is reversed compared to the bias asymmetry of the single electron transport regime[39]. In principle, these two regimes can be differentiated by varying $\Gamma_t$, since the conductance peaks increase linearly with $\Gamma_t$ in the single electron regime, and sublinearly in the resonant Andreev regime[39,64]. The experimental data shown in Fig. 3 display a marked asymmetry, consistent with our assumption of relaxation dominated transport where conductance asymmetry will follow the asymmetry of the underlying spectral function.

Next, we investigate the situations where the tip moves over the impurity along the surface, or towards the impurity as a function of tip-sample distance $\Delta d$. These situations are marked with blue and red arrows in Fig. 4a, respectively. In Fig. 4b, c we then plot sub-gap conductance as a function of level position, $\epsilon_0$, corresponding to the red/blue traces, assuming a linear dependence of $\epsilon_0$ with tip-sample distance. The agreement between our model and the data is good, both in terms of the energy dispersion and the asymmetry. Also, in both experiment and theory, additional conductance peaks at $eV = \pm(\Delta_t - E_{ig})$ are visible close to the singlet-doublet phase transition. We interpret these lines as the additional single electron processes shown in Fig. 4d, which arise from thermal population of the excited state close to the phase transition where $E_{ig} \lesssim k_B T$. The conductance peaks at $\Delta_t \pm E_{ig}$ meet at the point where the YSR states cross zero energy, signaling the change between singlet, and doublet ground states, and the asymmetry in intensity between the conductance peaks at $eV = \pm(\Delta_t + E_{ig})$ switches around.

The good agreement between this simple model (Fig. 4b, c) and the data presented in Fig. 2e and a, supports our interpretation that the tip exerts an effective gating of the impurity. We discuss alternative scenarios further in Supplementary Note 2, but the fact that our impurity is below the surface and the excellent agreement between the model and the data lead us to conclude that the gating scenario is most likely in the present case.

In summary, we have reported on the properties of energy symmetric in-gap states in FeTe$_{0.55}$Se$_{0.45}$ that can be tuned through the Fermi level. These states extend over a large (~10 nm) area around the center locations. Our data point towards a sub-surface magnetic impurity embedded in a low-density superfluid with large screening length that leads to YSR-like in-gap states. We propose a novel tip-gating mechanism for the dispersion and perform calculations within the single impurity Anderson model that show excellent agreement with the data. Such a mechanism could also play a role in previous experiments on elemental superconductors or heterostructures. How such states are related to the topological superconductivity in FeTe$_{0.55}$Se$_{0.45}$ remains an open question. Our work further shows

that one needs to be careful when interpreting zero-bias peaks in putative topological states, and junction resistance dependent experiments are a necessary—ideally combined with other techniques such as noise spectroscopy[65–67] (see also Supplementary Note 4), spin-polarized STM[68], or photon-assisted tunneling[69] will allow for better understanding. Independent of this, tunable impurity states like the one we report here could offer a platform to study quantum phase transitions, impurity scattering, and the screening behavior of superfluids.

## Methods

**Sample preparation.** The FeTe$_{0.55}$Se$_{0.45}$ single crystal samples were grown using the Bridgman method and show a critical temperature of $T_C = 14.5$ K. We cleave them at ultra-high vacuum ($P_{base} \sim 1 \times 10^{-10}$ mbar) and low temperature (~30 K) and immediately insert them to our pre-cooled STM (USM-1500, Unisoku Co., Ltd). The STM tips used in this work are mechanically sharpened Pt-Ir wires. They are Pb-coated by indenting them on a Pb(111) surface which was first cleaned by repetitive cycles of Ar sputtering ($P_{base} \sim 5 \times 10^{-5}$ mbar) followed by thermal annealing.

**Measurement.** Standard lock-in technique is employed for the tunnelling conductance measurements at 887 Hz. All measurements were performed at 2.2 K.

## Data availability
The data of this work are available from the corresponding author upon request.

## Code availability
The code used in this work is available from the corresponding author upon request.

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

## Acknowledgements

We acknowledge J. de Bruijckere, J.F. Ge, M.H. Fischer, P.J. Hirschfeld, D.K. Morr, P. Simon, J. Zaanen, and H.S.J. van der Zant for fruitful discussions. This work was supported by the European Research Council (ERC StG SpinMelt) and by the Netherlands Organization for Scientific Research (NWO/OCW), as part of the Frontiers of Nanoscience programme, as well as through a Vidi grant (680-47-536). G.G. is supported by the Office of Basic Energy Sciences, Materials Sciences and Engineering Division, US Department of Energy (DOE) under contract number de-sc0012704. B.M.A. acknowledges support from the Independent Research Fund Denmark grant number DFF-8021-

00047B. The Center for Quantum Devices is funded by the Danish National Research Foundation. D. Cho was supported by the National Research Foundation of Korea (NRF) grant funded by the Korea government (MSIT) (No. 2020R1C1C1007895 and 2017R1A5A1014862) and the Yonsei University Research Fund of 2019-22-0209. A.A. acknowledges the support of the Max Planck-POSTECH-Hsinchu Center for Complex Phase Materials, and financial support from the National Research Foundation (NRF) funded by the Ministry of Science of Korea (Grant No. 2016K1A4A01922028).

## Author contributions

D. Chatzopoulos, D. Cho, and K.M.B. performed the experiments and data analysis. D. Chatzopoulos, G.O.S., D.B., A.A., J.P., and B.M.A. performed the simulations. G.G. grew and characterized the samples. All authors contributed to the interpretation of the data and writing of the manuscript. M.P.A. supervised the project.

## Competing interests

The authors declare no competing interests.
