## [Peer Review File · Nature Communications]

REVIEWER COMMENTS

Reviewer #1 (Remarks to the Author):

The authors report on a magnetic impurity in an iron-based superconductor, which they have investigated by scanning tunneling microscopy and spectroscopy. They claim that the subgap state that they found, when moving over this impurity, is a Yu-Shiba-Rusinov (YSR) state. Changing the distance from the center of the impurity in x and y (in plane) as well as in d (tip-sample distance), they found that the energy of the YSR state changes. Since they say that the impurities are subsurface, they explain the energy shift by the tip position through an electric field induced by the tip at the subsurface impurity. They argue that the field can penetrate by about 0.5nm into the sample. The data are intriguing and the experiments seem to have been carried out carefully, however, I do not find the interpretation convincing and quite speculative. Therefore, I do not recommend this manuscript for publication. A more detailed account is given below:

1. The experimental results and the theoretical calculations seem quite disconnected. Although there is a resemblance in the evolution of the YSR energies, I am missing a quantitative connection between theory and experiment. At the very least, I would expect some estimate if the electric field generated by the tip can induce a level shift in the impurity such that the levels can shift as suggested by the calculation. Actual numbers would be helpful.
2. The screening length is estimated through the Thomas-Fermi screening length. Does this also hold in the superconducting state?
3. Following the arguments and the data, the tip has an influence on the impurity even 5nm away (see Fig. 1e). At this distance, a sizeable path from the tip to the impurity, probably several nm, would go through the sample, which makes me wonder how much of the electric field is screened away already. Should it not be possible to estimate the field at the impurity for different tip positions and then correlate this with the changes in the YSR energy in order to test the feasibility of the arguments?
4. In addition, the lateral influence and the vertical influence differ by an order of magnitude. This is quite surprising and the authors do not have a good explanation for this.
5. Line 52: It probably should say "The energy can also be tuned by..."
6. Line 72: Ref. 32 is not the first work to use superconducting tips for enhancing the energy resolution. There have been a number of papers before this (not necessarily a complete list): Physica C 404, 306–310 (2004); Eur. Phys. J. B 40, 483–488 (2004); Physica C 468, 537–542 (2008)
7. The authors never really specify the kind of impurity that they think they have. I admit that this will be difficult to do, but some reasonable argument could be given. In line 200, they call this a disorder site. What do they mean by this? Is it structural disorder? A foreign atom in a lattice site or in an interstitial site? How does the influence of the electric field narrow down the possibilities?
8. In line 222, the authors discuss the "opposite limit", when $\Gamma_t \gg \Gamma_r$. It is not correct to say that transport takes place via Andreev reflections. If anything this has to be called resonant Andreev reflections, because the YSR state replaces the virtual state for the Andreev reflection by a real state. It is well known that resonant tunneling is sequential, which changes its overall behavior, such that shot noise or photon-assisted tunneling experiments are not suitable for detecting the difference. A more evident and easier to measure signature of moving into the "opposite limit" is a transition from a linear to a sublinear regime as described in Ref. 37.

9. Despite the criticism, I am intrigued by the experimental data such that I want to offer an alternative explanation. To me, the results would make more sense if they come from an extended magnetic impurity, such as a cluster of a size of a few nm. This would be more like the case described in Ref. 45 with the magnetic island. I am not sure if it is possible to have, say, Fe clusters in the crystal from imperfect crystal growth or something like that. This could be tested from the magnetization behavior in a PPMS or so. In that case, I could imagine that the cluster is more susceptible to be polarized by the electric field of the tip that could then explain the tip-sample distance dependence. What is the density of these impurities at the surface? Do they all have the same extent?

Reviewer #2 (Remarks to the Author):

By using a superconducting tip, the authors have measured the tunneling spectra on iron based superconductors Fe(Te,Se). They report the discovery of energy symmetric in-gap states when measuring on top of some impurities (they regard them as magnetic impurity, probably the interstitial irons). The signal of these symmetric in-gap states becomes clear after using superconducting tip and the so-called deconvolution technique. The interesting point is that they find an "X" shape dispersion when they change the distance between the tip and the impurity. At a certain point the peak can appear exactly at zero bias, thus they interpret this dispersion by the tip induced gating effect by following the general picture of Anderson model for a single impurity in conventional superconductors. Thus concerning the "topological issue" recently appearing in this field, typically on this type of samples, their results remind that one needs to be careful when interpreting zero-bias peaks (ZBCP) in putative topological states. I think the data are nicely collected and carefully analyzed. The explanation is plausible. The conclusion is also quite important to provide an alternative interpretation for the zero-bias conductance peak at some impurity spots which have been regarded as the evidence of topological superconducting state. Thus I would be happy to recommend publication of this work in Nature Communications.

While since it is a piece of work which may overthrow the basic understanding of the impurity induced ZBCP as the origin of topological state in previous publications, it would be important for the authors to mention how is the statistics, for example how many impurities they have measured and how many of them show this type of "X" dispersion, and do they see impurities with other behaviors, such as robust ZBCP when moving the distance between the tip and impurity? After this message is added I suggest to accept the manuscript for publication.

Reviewer #3 (Remarks to the Author):

Chatzopoulos et al. have performed STM experiments and theoretical modeling to argue the in-gap bound states in the iron-based superconductor $\text{FeTe}_{0.55}\text{Se}_{0.45}$, which is a prime candidate for the topological superconductor. The early STM experiment reported the zero-energy bound state at the excess irons, and its relevance to the Majorana bound state was suggested. Later experiment showed that the bound-state energy depends on the tip-sample distance, probably because the tip varies the coupling between the excess iron and the superconductor surface through the force acting between the tip and the excess iron. In this manuscript, the authors investigate the Yu-Shiba-Rusinov states due to buried impurities that should be more rigid than the excess irons. Interestingly, even though it is unlikely that the impurity could be moved by the tip, the bound-state

energy depends on both lateral and vertical tip-sample distances. The authors propose a novel mechanism of the observed energy shift based on the tip-induced gating. Their theoretical model calculation has succeeded in reproducing the qualitative features of the observations. The experiments have been carefully performed and the idea of the tip-induced gating is unique. The proposed theoretical model is simple and reasonable. As the authors mention, it is not clear the relation between the present results and the topological nature. Having said that, the proposal of the manuscript is important for the future Majorana search using magnetic impurities and should apply not only for $\text{FeTe}_{0.55}\text{Se}_{0.45}$ but also for other low-carrier superconductors. I recommend the publication of this manuscript in Nature Communications as it is.

Point-by-point reply to the reviewers

We thank all the reviewers for their helpful feedback, we believe it has made our paper much stronger.

We are grateful that the reviewers point out the novelty and the ground-breaking significance of our approach and results, and we thank Reviewer #2 and #3 for endorsing publication in Nature Communications. We are also grateful for the frank remarks on where clarification and improvement are needed; we are confident that we can clarify or rebut all issues.

All changes are marked in blue in the manuscript.

With kind regards,

Milan Allan on behalf of the authors

REVIEWER COMMENTS

Reviewer #1 (Remarks to the Author):

The authors report on a magnetic impurity in an iron-based superconductor, which they have investigated by scanning tunneling microscopy and spectroscopy. They claim that the subgap state that they found, when moving over this impurity, is a Yu-Shiba-Rusinov (YSR) state. Changing the distance from the center of the impurity in x and y (in plane) as well as in d (tip-sample distance), they found that the energy of the YSR state changes. Since they say that the impurities are subsurface, they explain the energy shift by the tip position through an electric field induced by the tip at the subsurface impurity. They argue that the field can penetrate by about 0.5nm into the sample. The data are intriguing and the experiments seem to have been carried out carefully, however, I do not find the interpretation convincing and quite speculative. Therefore, I do not recommend this manuscript for publication. A more detailed account is given below:

We thank the reviewer for taking the time to thoroughly read our manuscript and for the comments and suggestions.

1. The experimental results and the theoretical calculations seem quite disconnected. Although there is a resemblance in the evolution of the YSR energies, I am missing a quantitative connection between theory and experiment. At the very least, I would expect some estimate if the electric field generated by the tip can induce a level shift in the impurity such that the levels can shift as suggested by the calculation. Actual numbers would be helpful.

We respectfully disagree that experimental results and theoretical calculations are disconnected. We took great care in displaying both qualitative and quantitative comparisons between theory and experiments. Figures 3 and 4 are designed to make a comparison for the reader as easy as possible. As it is visible from these figures, the qualitative agreement is good, but also the quantitative agreement, e.g. when looking at the energy and its dispersion.

We do agree with the referee that the paper would benefit from both more numbers in the text, and from an estimate of the electric field, and we thank her/him for the suggestion.

We start with a comparison of the magnitude of the electric field of the tip with the charging energy of the impurity. According to the simulations in Figure 4a, in order to observe a dispersion of the YSR state from the gap edge $\sim\Delta_s$ to zero energy one needs to shift the energy level ε by half the charging energy U . An accurate estimation of the charging energy of the observed impurity is challenging since its origin is unknown. However, a simple estimation can be made if we take it to be a charged sphere with radius R . Then its charging energy would be given by $U=e^2/(8\pi\epsilon_0R)$. Taking $R\sim 1-3\text{nm}$ we estimate

$U/2 \sim 100\text{-}350\text{meV}$. This is smaller than the estimated electric field penetration in our sample due to the work-function difference between tip and sample is $\sim 1\text{eV}$ (see supplementary). Hence the gating scenario is possible.

We can also follow a different approach. To be able to gate the YSR state through zero energy, we imagine a scenario like the Numerical Renormalization Group calculations in Bauer & Hewson (Ref 59, Figs. 9 and 11 shown below) where $U=5\pi\Gamma$ (lower left figure below). We choose this red curve since at $\xi_d=0$ it has YSR at nearly the same fraction as ours ($\sim 0.6\Delta_s$). This fig. has $\pi\Gamma=20\Delta$, meaning that $\pi\Gamma=40\text{meV}$, say. This means that $U=200\text{meV}$. To move through zero energy in the lower left plot, the level position should change by approximately $0.4U=80\text{meV}$. This should correspond to the screened value of the work function difference of 1eV , which sounds reasonable too. Last but not least a charging energy of 200meV corresponds to a charged sphere with radius of $\sim 3\text{nm}$, that is comparable to the size of the impurity feature that we observe in the LDOS maps of Figure 2.

In response to the Reviewer's comments, we now added an extra section (VI) in our supplementary material titled: "Estimation of the charging energy U of the impurity" where we analyze the reasoning above.

[Redacted]

2. The screening length is estimated through the Thomas-Fermi screening length. Does this also hold in the superconducting state?

This is an interesting question and we thank the reviewer for bringing up this point. The Thomas-Fermi screening length is usually applicable in the normal state where normal carriers act as screeners for

electric fields. The Thomas-Fermi screening length is also a good approximation even in our case of a low electron density superconductor. When temperature is below the superconducting critical temperature one expects no or only low order corrections of the order of Δ_s^2/E_F^2 to the Thomas-Fermi screening length (Phys. Rev. B 70, 226503). This is because all filled states are integrated over when calculating the screening length, and then a small gap near the Fermi level is essentially irrelevant.

We note that this topic is still debated in the literature. There is some rather controversial work that claims that the screening of electric fields is governed by the London penetration depth, which in Fe(Se,Te) is hundreds of nanometers (including Phys. Rev. B 68, 184502 (2003), Physics Letters A 372 (2008) 3289–3291, Chinese Phys. B 27 057401 (2018)).

We thus argue that the Thomas-Fermi screening length is appropriate in our case and that our estimation based on the Thomas-Fermi approximation is accurate and predicts a screening length that is still large enough to reach the impurity in the subsequent layer. In order to discuss this interesting aspect, we now added a few sentences in the second paragraph of section E.

3. Following the arguments and the data, the tip has an influence on the impurity even 5nm away (see Fig. 1e). At this distance, a sizeable path from the tip to the impurity, probably several nm, would go through the sample, which makes me wonder how much of the electric field is screened away already. Should it not be possible to estimate the field at the impurity for different tip positions and then correlate this with the changes in the YSR energy in order to test the feasibility of the arguments?

We are not sure if we understand the Reviewer correctly. If she/he refers to the electric field penetration, please refer to the reply to point 4. If the Reviewer refers to the extension of the state, then we note that the extension of the YSR wavefunction in conventional superconductors is up to the coherence length ξ (see for example ref 16), which in the case of Fe(Te,Se) is $\sim 3\text{nm}-10\text{nm}$ (Sci. Adv. 6 (9), eaay0443, <https://arxiv.org/abs/0911.4758v1>). Hence, we do not necessarily need a sizeable path from the tip position to the impurity center. As long as the extended feature of the YSR state is affected by the electric field lines of the tip locally there should be an influence on the YSR energy.

4. In addition, the lateral influence and the vertical influence differ by an order of magnitude. This is quite surprising and the authors do not have a good explanation for this.

This is a good point that can be explained. The tunneling will always be from the last atom to the surface right underneath, as the tunneling probability is exponential. The electric field, on the other hand, decays algebraically. In addition, we must keep in mind that while the end of the tip is atomically sharp, the tip-radius of the apex is usually 10-100nm. In our case, we estimate it to be roughly 20nm, although this is a very rough estimate, as the preparation (indentation) on the Pb surface changes the tip shape. As Figure R1 below shows, a difference in z has much more influence on the electric field at the impurity location than a difference in x . This is first because of the sheer size of the tip radius compared to the depth of the impurity, but also because the lateral movement causes much of the potential drop to happen in vacuum, where it is not screened.

Figure R1: Tunneling junction schematics (in scale). Two spherical shaped tips (radius is 20 nm) that are 5 nm apart are shown. An impurity resides below the surface (atomic planes are drawn as blue lines separated by 6 Å).

Figure R2: Top left: tunneling distance is 1 Å. Top right: tunneling distance is 2 Å. Right: Tunneling distance is 1 Å. and 5nm on the right. Tip radius is 15nm in all cases. We note that this is just a pedagogical model to illustrate our point, using a metallic tip of radius 15nm and a semiconductor sample with a screening length similar to ours.

We further note that the fact that the vertical and lateral influence differ by an order of magnitude is an extra reason to exclude other scenarios e.g. tip-force (PRL 121,196803 (2018)) in which the YSR state can only be tuned by changing the tip-sample distance (Δd).

The discussion above already indicates that estimating absolute numbers for the gating is challenging because the penetration of electronic fields on the surface of superconductors at such short distances is not well understood. The estimate that we describe in our reply to point 2 (and inserted to the manuscript in section E) is a lower bound that demonstrates the applicability of our method. Finally, in order to discuss the tip-shape effects we updated the Supplementary Figure S5 with a schematic like figure R1 and added a paragraph in supplementary section V, while being careful not to overstate the precision of our estimates.

5. Line 52: It probably should say "The energy can also be tuned by..."

We thank the reviewer for pointing this typo, which we corrected.

6. Line 72: Ref. 32 is not the first work to use superconducting tips for enhancing the energy resolution. There have been a number of papers before this (not necessarily a complete list): Physica C 404, 306–310 (2004); Eur. Phys. J. B 40, 483–488 (2004); Physica C 468, 537–542 (2008)

We sincerely apologize for the oversight. We added the following citations to the main text in order to give proper credit to these early works: Appl. Phys. Lett. 73, 2992 (1998), Eur. Phys. J. B 40, 483–488 (2004).

7. The authors never really specify the kind of impurity that they think they have. I admit that this will be difficult to do, but some reasonable argument could be given. In line 200, they call this a disorder site. What do they mean by this? Is it structural disorder? A foreign atom in a lattice site or in an interstitial site? How does the influence of the electric field narrow down the possibilities?

As the reviewer points out determining the kind of the impurity at hand is not an easy task. Our paper shows that we can still extract much understanding from our data, as confirmed by Reviewr #2 and #3. Importantly, even if one knows what kind of impurity is in the sample, the interpretation does not become trivial. As discussed e.g. in PRL 105, 157004 (2010) for dopants, or for magnetic impurities in the cuprates, a putative magnetic impurity (say, iron) does not necessary stay magnetic if it is in a superconducting host. Thus, even if the impurity is known, one often still needs to carefully think along the arguments that we present on the conclusions section.

Still, we are able to make quite some constraints on the nature of the impurity. At first one might think that the obvious choice is magnetic excess Fe atoms or Fe clusters. However, we believe that such a scenario is less probable here for the following reasons (see also reply to comment 9). First of all, we have not observed excess Fe atoms or Fe clusters on the surface. In addition, to prevent surface reconstruction and contamination we cleave our samples in ultrahigh vacuum (1×10^{-10} mbar) at low temperature ($T \approx 30$ K) and are immediately mounted in the pre-cooled STM head ($T \approx 2.2$ K). These two aspects are now emphasized on lines 61 and 80-82. We agree that in line 200 the term disorder site is not well defined. We do not observe signatures of structural disorder, foreign atom substitution or interstitial disorder. We now changed the wording to "impurity site".

8. In line 222, the authors discuss the "opposite limit", when $\Gamma_t \gg \Gamma_r$. It is not correct to say that transport takes place via Andreev reflections. If anything this has to be called resonant Andreev reflections, because the YSR state replaces the virtual state for the Andreev reflection by a real state. It is well known that resonant tunneling is sequential, which changes its overall behavior, such that shot noise or photon-assisted tunneling experiments are not suitable for detecting the difference. A more evident and easier to measure signature of moving into the "opposite limit" is a transition from a linear to a sublinear regime as described in Ref. 37.

We agree with the reviewer that resonant Andreev reflection is better terminology, which we have now adopted in the main text. Furthermore, we accidentally wrote that these resonant Andreev reflections lead to a conductance which is symmetric under reversal of bias voltage. This should have been the *current* which is symmetric, while the conductance peaks remain asymmetric with an asymmetry which is opposite to that of the single electron transport regime. Speculations on shot-noise and photo-assisted tunneling have been removed and replaced by characterization using the linear to sublinear shift in Γ_t dependence, as suggested by the reviewer.

9. Despite the criticism, I am intrigued by the experimental data such that I want to offer an alternative explanation. To me, the results would make more sense if they come from an extended magnetic impurity, such as a cluster of a size of a few nm. This would be more like the case described in Ref. 45 with the magnetic island. I am not sure if it is possible to have, say, Fe clusters in the crystal from imperfect crystal growth or something like that. This could be tested from the magnetization behavior in a PPMS or so. In that case, I could imagine that the cluster is more susceptible to be polarized by the electric field of the tip that could then explain the tip-sample distance dependence. What is the density of these impurities at the surface? Do they all have the same extent?

We are very happy to see that the reviewer is intrigued by the data as we are. There are several candidates attributed to the in-gap states. The reviewer makes a good suggestion of a magnetic island below the surface, as the reviewer pointed out. There are several reasons why we ruled out the scenario. First of all, we've never seen the excess Fe cluster (or single atom) on the surface. To exclude the migration of Fe atoms and the surface reconstruction, we cleaved our samples at low temperature (~ 30 K) and transferred them to the precooled STM head (2.3 K). Second, the spatial distribution of our in-gap states is clearly different from the previous STS studies of magnetic islands on superconductors. Their zero-energy bound states are localized at the edge of the islands and the nonzero in-gap states show two concentric circles in the conductance maps (see below). However, our zero-bound state is always localized at the atomic site where the circle of nonzero-energy bound states is centered.

Nature Communications, 8, 2040 (2017)

Third, the sharp transition at 14.0 ± 0.5 K guarantees an absence of the magnetic impurities in our samples (Eur. Phys. J. B 79, 289–299 (2011)). This has been thoroughly investigated by growing samples with and without excess iron which can be diminished by annealing.

The defect density is very low (see also our reply to Reviewer #2), but all of them shows similar extent around 5 nm. We added a note to the main text in section B and a supplementary section VII about statistics.

Reviewer #2 (Remarks to the Author):

By using a superconducting tip, the authors have measured the tunneling spectra on iron based superconductors Fe(Te,Se). They report the discovery of energy symmetric in-gap states when measuring on top of some impurities (they regard them as magnetic impurity, probably the interstitial irons). The signal of these symmetric in-gap states becomes clear after using superconducting tip and the so-called deconvolution technique. The interesting point is that they find an “X” shape dispersion when they change the distance between the tip and the impurity. At a certain point the peak can appear exactly at zero bias, thus they interpret this dispersion by the tip induced gating effect by following the general picture of Anderson model for a single impurity in conventional superconductors. Thus concerning the “topological issue” recently appearing in this field, typically on this type of samples, their results remind that one needs to be careful when interpreting zero-bias peaks (ZBCP) in putative topological states. I think the data are nicely collected and carefully analyzed. The explanation is plausible. The conclusion is also quite important to provide an alternative interpretation for the zero-bias conductance peak at some impurity spots which have been regarded as the evidence of topological superconducting state. Thus, I would be happy to recommend publication of this work in Nature Communications.

We thank the reviewer for recommending publication to Nature Communications.

While since it is a piece of work which may overthrow the basic understanding of the impurity induced ZBCP as the origin of topological state in previous publications, it would be important for the authors to mention how is the statistics, for example how many impurities they have measured and how many of them show this type of “X” dispersion, and do they see impurities with other behaviors, such as robust ZBCP when moving the distance between the tip and impurity? After this message is added I suggest to accept the manuscript for publication.

We gladly clarify this important point. On this sample, we investigated 5 ring-shaped in-gap impurity features in a 45x45 nm² field-of-view. The one shown in the main text shows the X-shaped profile while moving both Δd and Δr . The remaining 4 impurities (no topographic signature is observed) have slightly smaller spatial extension and disperse spatially in a similar fashion to the one shown in the main text, however a cross at the Fermi level is not observed at $\Delta d = -0.7 \text{ \AA}$. We did not measure all 5 impurities for smaller tip-sample distances but we expect similar X-shaped profiles. The reason why the 5 impurities have different spatial extension and dispersion could be due to their location inside the crystal. We can imagine that an impurity that is deeper in the crystal would be less prone to the influence of the electric field of the tip. Similarly, on a second sample we found 2 additional impurity features extending over $\sim 4 \text{ nm}$. Thus, this kind of impurities is rather sparse, but show reproducible phenomenology.

We do not observe impurities with robust zero-bias peaks that do not change when changing the tip-sample distance.

We added a message about statistics in the supplementary (section VII) and the main text (lines 104-106).

Reviewer #3 (Remarks to the Author):

Chatzopoulos et al. have performed STM experiments and theoretical modeling to argue the in-gap bound states in the iron-based superconductor $\text{FeTe}_{0.55}\text{Se}_{0.45}$, which is a prime candidate for the topological superconductor. The early STM experiment reported the zero-energy bound state at the excess irons, and its relevance to the Majorana bound state was suggested. Later experiment showed that the bound-state energy depends on the tip-sample distance, probably because the tip varies the coupling between the excess iron and the superconductor surface through the force acting between the tip and the excess iron. In this manuscript, the authors investigate the Yu-Shiba-Rusinov states due to buried impurities that should be more rigid than the excess irons. Interestingly, even though it is unlikely that the impurity could be moved by the tip, the bound-state energy depends on both lateral and vertical tip-sample distances. The authors propose a novel mechanism of the observed energy shift based on the tip-induced gating. Their theoretical model calculation has succeeded in reproducing the qualitative features of the observations. The experiments have been carefully performed and the idea of the tip-induced gating is unique. The proposed theoretical model is simple and reasonable. As the authors mention, it is not clear the relation between the present results and the topological nature. Having said that, the proposal of the manuscript is important for the future Majorana search using magnetic impurities and should apply not only for $\text{FeTe}_{0.55}\text{Se}_{0.45}$ but also for other low-carrier superconductors. I recommend the publication of this manuscript in Nature Communications as it is.

We thank the reviewer for sharing our excitement about the future implications of our study and for endorsing publication in Nature Communications!

REVIEWER COMMENTS

Reviewer #1 (Remarks to the Author):

The authors have addressed my points for the most part. However, there are still some concerns that remain. My main point still is that I would like to see a more quantitative connection between the experimental data and the corresponding calculations. The authors provide fragments of arguments throughout the manuscript, which makes it very difficult for the reader to follow even in the new version. If this paper should be recommended for publication in Nature Communications, I expect a more transparent discussion of the proposed mechanism and a more complete presentation of all available information. I will outline this in the following:

1. Concerning my previous point about providing actual numbers to support their claim, the authors do not provide a satisfactory answer. They calculate the charging energy of a sphere with one electron having a radius of 1 to 3nm, which sounds reasonable when comparing with the dI/dV maps. However, I would like to see a more detailed estimate of the energy shift induced by the applied electric field. They simply state that the energy difference between the work functions is about 1eV, but this does not tell me anything about the change in the potential at the impurity when changing the distance of the tip. They estimate the Thomas-Fermi wavelength, so to me the next logical step would be to assume that the electric field decays exponentially into the bulk. The electric field at the surface can be estimated by the work function difference and the tip sample distance. Then the change in potential at the impurity could be estimated from the exponential decay of the electric field (i.e. some kind of band bending). The tricky part is to estimate the distance between tip and sample, but this could be done, for example, by using the normal state conductance (which they do not provide in the manuscript) and setting the conductance quantum as a zero distance. Overall, I am not convinced that the provided numbers fit as well as the authors suggest.

The estimate using the paper by Bauer et al (which is not Ref. 59 in the current version of the manuscript) is not convincing. The entire data in Fig. 9 was calculated assuming a symmetric case (i.e. $\chi_d = 0$, $\epsilon_d = -U/2$), which is clearly not the case in the present work as their argument is centered around a changing ϵ_d . Further, their calculation implies that $\Gamma = 13\text{meV}$, which seems to be rather small for an embedded impurity.

I think the authors should not use controversies in the literature to be vague about their arguments that support their findings. Instead I think they should take a stand with the available information (and I think the necessary information is there) and provide a comprehensible analysis. I think, if the authors themselves state in the manuscript that "we are not able to exclude an alternative scenario, in which the impurity-substrate coupling [...] depends monotonically on the tip distance," they are not completely convinced by their interpretation, so that they should make a much stronger point, especially if they want to publish in a high profile journal such as Nature Communications. Incidentally, can the authors exclude the scenario that the impurity actually looks like any other atom in the lattice in the topography?

2. In addition, the authors state that the "conductance asymmetry will follow the asymmetry of the underlying spectral function." Since the asymmetry is a hallmark of a shifting of the energy level, a more detailed analysis and comparison of the the asymmetry in theory and experiment would be desirable.

Reviewer #2 (Remarks to the Author):

I think the authors have addressed very well of my concern about the statistics. They measured 5-ring shaped impurities and find all show similar behavior, this makes me more confident about the final conclusion. I believe now the work is acceptable for publication.

POINT-BY-POINT REPLY TO REVIEWER COMMENTS

Reviewer #1 (Remarks to the Author):

The authors have addressed my points for the most part. However, there are still some concerns that remain. My main point still is that I would like to see a more quantitative connection between the experimental data and the corresponding calculations. The authors provide fragments of arguments throughout the manuscript, which makes it very difficult for the reader to follow even in the new version. If this paper should be recommended for publication in *Nature Communications*, I expect a more transparent discussion of the proposed mechanism and a more complete presentation of all available information. I will outline this in the following:

We thank Reviewer #1 for assessing our revised manuscript. We are glad that we could address Reviewer #1's points for the most part. We give a more transparent discussion of the proposed mechanism, including more detailed estimates on the tip height and field penetration requested by the review, both in the manuscript and below. We also updated Figure 4 for clarity and to reflect the improved estimates.

However, we note that Fe(Te,Se) is a material that is highly unusual: it has an extremely low Fermi energy, shows signs of being a topological superconductor, and hosts a highly disordered superfluid. This is why the material is of such interest to many physicists. In our paper, we do not want to give the impression that estimates that are based on text-book metals are adequate for Fe(Te,Se). Nevertheless, we appreciate the reviewer's suggestions, and improved the manuscript based on the reviewer's comments.

We also want to politely mention that the question of how approximative the estimates should be is, in our opinion, more of a (very interesting) side discussion that is not key to our paper. In our paper, we find evidence for gating as the mechanism for a novel YSR state. This is novel and different to the traditionally discussed force-mechanism. Note that the papers that found the force-mechanism for YSR, which were deservedly published in *Nature Physics*, *PRL*, etc, could not precisely estimate the exact force between molecule and tip. But as readers and scientists, we appreciate the progress they reported.

Still, given Reviewer #1's comments, we do want to make sure that our discussion is complete and transparent. Apart from the detailed reply below, we changed the text in the following (all changes from revision 2 are indicated in red, those of revision 1 in blue):

1. Lines 189-194 include a discussion about the potential shift on the impurity when the tip-sample distance is varied. In connection to that, we expanded supplementary note 4 with a detailed estimation of the change in the potential on the impurity as the tip-sample distance is varied. We also updated the estimation of the charging energy, taking into account the dielectric constant of the sample.
2. We updated Fig 4, panels a,b and c. The simulations are now performed with the improved estimates for the parameters U and Γ_s , as explained below.
3. We noticed that the y-axis in Figs. 4a-c were inverted and that in the caption, we incorrectly stated that the color-scale is the same for panels b and c. In addition, in Fig 3a the x-axis values were not correct. These are now corrected in the current version.

4. We deleted the sentence “At this point, we are not able to exclude an alternative scenario, in which the impurity-substrate.. the tip distance [49].”, since we discuss the reason we discard this scenario in the supplementary method 2.

5. We added a few references to improve the clarity. In the main text: *Symmetry* 2020, 12, 1402. In the supplementary file: PRB 95, 235141 (2017), *Sci. Adv.* 4, 10.1126/sciadv.aao2682 (2018). Moreover we replaced citation 15 (arXiv) preprint with its published version.

6. We changed the format of our manuscript and supplementary information such that it complies with the Nature Communications format.

1. Concerning my previous point about providing actual numbers to support their claim, the authors do not provide a satisfactory answer. They calculate the charging energy of a sphere with one electron having a radius of 1 to 3nm, which sounds reasonable when comparing with the dI/dV maps. However, I would like to see a more detailed estimate of the energy shift induced by the applied electric field. They simply state that the energy difference between the work functions is about 1eV, but this does not tell me anything about the change in the potential at the impurity when changing the distance of the tip. They estimate the Thomas-Fermi wavelength, so to me the next logical step would be to assume that the electric field decays exponentially into the bulk. The electric field at the surface can be estimated by the work function difference and the tip sample distance. Then the change in potential at the impurity could be estimated from the exponential decay of the electric field (i.e. some kind of band bending). The tricky part is to estimate the distance between tip and sample, but this could be done, for example, by using the normal state conductance (which they do not provide in the manuscript) and setting the conductance quantum as a zero distance. Overall, I am not convinced that the provided numbers fit as well as the authors suggest.

As we stated in our previous reply to the reviewer, making an accurate estimate of the potential shift in the impurity when changing the tip-sample distance is challenging. In the following we will try to estimate the aforementioned potential shift using methods that have been applied to semiconductors or correlated systems (band bending). However, as we also stated before, we don't want to give the impression that Fe(Te,Se) can be described adequately as semiconductor or other text-book materials. We rather want to show that even in this simple approach the gating scenario is applicable in order to explain our experimental findings.

Following the Reviewer, we estimate the tip sample distance using the $G(h)$ spectroscopy. The figure below shows a conductance (G) versus tip vertical displacement (h). We fit the datapoints (black dots) with an exponential function (blue curve) and extrapolate to the conductance quantum to find the sample position.

Figure R1: Conductance versus tip displacement curve measured on the same sample as the data in Figure 2 of the main text. Black dots are measured points which are fitted with an exponential and extrapolated to the conductance quantum G_0 . Red dots correspond to the conductance versus tip displacement data presented in Fig. 3 of the main text.

We then find the tip-sample distance for the data shown in Fig. 3 by using the fitting function to convert the setup normal state conductance to tip displacement (red dots in the figure above). This gives us a minimum distance of 2.9 Å and a maximum distance of 5.1 Å between tip and sample.

For estimating the potential drop on the surface of the sample (ϕ_{BB}) we may use the following formula as derived for tip-induced band bending phenomena, previously reported for semiconductors and correlated electron systems (PRB 95, 235141 (2017)) :

$$\phi_{BB} = \frac{1}{1 + \varepsilon \frac{h}{r}} (V_b - W)$$

here ε is the dielectric constant of the sample, h the tip-sample distance, r the tip radius and V_b the applied bias between tip and sample (for a schematic see the picture below). The formula given above gives the potential at the surface of the sample (point A) in the schematics.

[Redacted]

Figure: Schematic of the tip and sample.
Reproduced from PRB 95, 235141 (2017)

For making an accurate estimate of the potential at point A we choose the values for the parameters as realistic as possible. We will choose $r = 20$ nm, $W = 1$ V and we will set $V_b = 5$ mV (set-up bias in Fig 3). Next, we estimate the dielectric constant of the sample to be of the similar order as FeSe $\varepsilon \sim 15$ (Sci Adv 4(3) eao2682).

We then plot the ϕ_{BB} as a function of the tip-sample distance for the parameters described above.

Potential drop on the sample surface as a function of the tip-sample distance. Red vertical lines indicate the range of the movement of our tip.

From this plot we see that the potential change is ~ 0.1 V on the surface. Our next step is to calculate the exponential decay of the potential inside the bulk. Since we employed the Thomas-Fermi approximation:

$$V(r_1) = V_0 e^{-r_1/\lambda_{TF}}$$

where r_1 is the distance from the sample surface to the impurity location, V_0 is the potential change that we estimated before and λ_{TF} the Thomas-Fermi screening length ~ 0.5 nm. If we assume that $r_1 = 0.25$ nm (impurity resides midway between top and next layer) then we have that the change in potential felt by the impurity for the tip movement in our experiment is 60 mV. We see that this is comparable to the lower bound of $U/2=100-350$ mV that is required (according to our model) in order to shift the impurity states from zero bias, close to the gap edge. The agreement becomes even better if we correct the charging energy U for the dielectric constant ϵ . That is $U=e^2/(8\pi\epsilon\epsilon_0 R)$. In that case U becomes 15-50 meV, which is in the range of the estimated potential change on the impurity (for more details also see the next answer).

All these considerations show us that the gating scenario is possible. We now included this discussion in lines 189-194 of the main text and the supplementary note 4.

The estimate using the paper by Bauer et al (which is not Ref. 59 in the current version of the manuscript) is not convincing. The entire data in Fig. 9 was calculated assuming a symmetric case (i.e. $\xi_d = 0$, $\epsilon_d = -U/2$), which is clearly not the case in the present work as their argument is centered around a changing ϵ_d . Further, their calculation implies that $\Gamma = 13$ meV, which seems to be rather small for an embedded impurity.

We agree that Fig. 9 of Bauer et al. is shown in the case of $\epsilon_0 = -U/2$, whereas we consider also different values of ϵ_0 . Our reference to the Bauer result in our previous reply of course doesn't say anything quantitatively about the magnitudes of Γ_s nor U , and was merely meant as one potentially consistent scenario. The same could be achieved, however, for other values of Γ_s and U . All we know for sure, is that we can move the level enough to observe the YSR state cross zero energy. In the supplement we point out that this implies that T_K is smaller than $0.3\Delta_s$, or more accurately in terms of Bauer's Fig. 12,

that Γ_s/U lies below the top point of the doublet phase at $\varepsilon_0=-U/2$. If this was not the case, one could no longer tune the level through zero energy (from singlet to doublet) by changing ε_0 .

In the following we estimate for a range of realistic parameters for our system, the values of Γ_s that allow to see the crossing based on Bauer's paper. For our system we expect that $U \approx 20$ meV (according to the latest estimation), and $\Delta_s \approx 1$ meV. Assuming the red curve in Bauer's Fig. 12, we have $\Gamma_s = \Delta_s / (0.3\pi) \approx 1$ meV. In addition, $U / (\pi\Gamma_s) \approx 6$ (i.e. $\Gamma_s/U \approx 0.05$, allowing the singlet/doublet transition) and from Bauer's eq. 19 thereby $T_K / \Delta_s \approx 0.0008$. On the other hand, for $U \approx 40$ meV then $\Gamma_s = \Delta_s / (0.1\pi) \approx 3.2$ meV would make $T_K / \Delta_s \approx 0.02$, which still allows the singlet/doublet transition, whereas $\Gamma_s = \Delta_s / (0.05\pi) \approx 6.4$ meV would correspond to the blue line in Bauer's Fig.12, but now with $T_K / \Delta_s \approx 0.4$ and $\Gamma_s/U \approx 0.16$, which is no longer consistent with observing the singlet/doublet transition. This is roughly the freedom provided by the single-orbital Anderson impurity in a superconductor.

Even though this is a side discussion, in our opinion, we want to make sure that our simulations in Fig. 4 reflect the present experiment case as much as possible. Given our new estimation for $U \approx 20$ meV and $\Delta_s \approx 1$ meV we updated the simulations in Fig. 4 with these parameters. For the value of Γ_s we choose the value of 3 meV which shows a crossing of the YSR states at zero bias, matching our observations. This choice of parameters corresponds roughly to the blue curve of Bauer's Fig. 12, allowing the singlet/doublet transition. However, we point out that we are showing results from a zero-bandwidth approximation (ZBA) to the full Anderson model. Therefore, the value of Γ_s applied within ZBA does not correspond exactly to the Γ_s used in Bauer's numerical renormalization group calculations.

Finally, from Fig 4 we now observe that we need to shift the energy on the impurity by $0.1U \sim 2$ meV in order to have the YSR states dispersing from the gap edge to zero bias. According to our rough estimations from the previous point we can induce a change of ~ 60 meV thus the gating scenario is possible.

I think the authors should not use controversies in the literature to be vague about their arguments that support their findings. Instead I think they should take a stand with the available information (and I think the necessary information is there) and provide a comprehensible analysis. I think, if the authors themselves state in the manuscript that "we are not able to exclude an alternative scenario, in which the impurity-substrate coupling [...] depends monotonically on the tip distance," they are not completely convinced by their interpretation, so that they should make a much stronger point, especially if they want to publish in a high profile journal such as Nature Communications.

We respectfully note that we cannot make decade long discussion and controversy, which is not directly related to our findings, go away in our paper. In our paper, we present our data, and discuss our findings in context of the literature. We have discussed the point of the screening length (albeit it is not central to our paper) fairly and transparently.

Incidentally, can the authors exclude the scenario that the impurity actually looks like any other atom in the lattice in the topography?

The important point is that there *is* an impurity state, and we have been completely honest in our discussion about what we know and what we do not know about this state. And our results are independent of the chemical nature of the impurity, as we discuss in detail in the manuscript. We further note that chemical differences on the surface are generally visible in STM, as the Te/Se in our material or the Bi/Pb in the cuprates show.

2. In addition, the authors state that the "conductance asymmetry will follow the asymmetry of the underlying spectral function." Since the asymmetry is a hallmark of a shifting of the energy level, a more detailed analysis and comparison of the asymmetry in theory and experiment would be desirable.

In Fe(Te,Se) in-gap states that are induced by impurities have been shown to be asymmetric in terms of their spectral weight due to particle-hole asymmetry that is present in the material [T. Machida *et al.* Nature Materials 18, 811–815 (2019)]. This is also visible in our data (see for example Fig. 1f and 1g). On the other hand, the superconducting single impurity Anderson model that we introduced in the main text, exhibits spectral weight asymmetry for $\epsilon_0 \neq -0.5U$.

Moreover, the feature of the asymmetry changing sign after the in-gap states cross at zero bias is well captured qualitatively by our simulations (Fig 4) supporting the gating scenario, as we already suggest in the manuscript (lines 242-244). This asymmetry change signals a change between singlet, and doublet ground states, associated with the YSR states cross at zero-bias (lines 254-257, see also Supplementary Method 2).

The asymmetry in spectral weight is indeed a hallmark as the reviewer points out. We believe that we have discussed this adequately in the manuscript and the Supplementary Information.

Reviewer #2 (Remarks to the Author):

I think the authors have addressed very well of my concern about the statistics. They measured 5-ring shaped impurities and find all show similar behavior, this makes me more confident about the final conclusion. I believe now the work is acceptable for publication.

We thank reviewer #2 for endorsing publication in Nature Communications.